# Development of Clinical Prediction Score for Chemotherapy Response in Advanced Non-Small Cell Lung Cancer Patients

**DOI:** 10.3390/healthcare11030293

**Published:** 2023-01-18

**Authors:** Chawalit Chayangsu, Jiraporn Khorana, Chaiyut Charoentum, Virote Sriuranpong, Jayanton Patumanond, Apichat Tantraworasin

**Affiliations:** 1Department of Internal Medicine, Surin Hospital, Institute of Medicine, Suranaree University of Technology, Surin 32000, Thailand; 2Division of Pediatric Surgery, Department of Surgery, Faculty of Medicine, Chiang Mai University, Chiang Mai 50200, Thailand; 3Clinical Epidemiology and Clinical Statistics Center, Faculty of Medicine, Chiang Mai University, Chiang Mai 50200, Thailand; 4Department of Internal Medicine, Faculty of Medicine, Chiang Mai University, Chiang Mai 50200, Thailand; 5Department of Internal Medicine, Faculty of Medicine, Chulalongkorn University & The King Chulalongkorn Memorial Hospital, Bangkok 10330, Thailand; 6Clinical Surgical Research Center, Chiang Mai University, Chiang Mai 50200, Thailand

**Keywords:** clinical prediction rule, chemotherapy, limited resources, advanced lung cancer, decision making

## Abstract

The outcomes of advanced non-small cell lung cancer (NSCLC) patients have been significantly improved with novel therapies, such as tyrosine kinase inhibitors and immune checkpoint inhibitors. However, in resource-limited countries, platinum-doublet chemotherapy is mainly used as a first-line treatment. We investigate clinical parameters to predict the response after chemotherapy, which may be useful for patient selection. A clinical prediction score (CPS) was developed, based on data from a retrospective cohort study of unresectable stage IIIB or IV NSCLC patients who were treated with platinum-doublet chemotherapy in the first-line setting with at least two cycles and an evaluated response by RECIST 1.1 at Surin Hospital Cancer Center, Thailand, between July 2014 and December 2018. The clinical parameters in the prediction model were derived by risk regression analysis. There were 117 responders (CR or PR) and 90 non-responders (SD or PD). The clinical prediction score was developed by six clinical parameters including gender, age, smoking status, ECOG, pre-treatment albumin, and histologic subtype. The AuROC of the model was 0.71 (95% CI 0.63–0.78). The internal validation was performed using a bootstrap technique and showed a consistent AuROC of 0.66 (95% CI 0.59–0.72). The prediction score ranged from 0–13, with a score of 0–8 meaning a low probability (PPV = 50%) and a score of 8.5–13 meaning a high probability (PPV = 83.7%) for chemotherapy response. Advanced NSCLC patients who cannot access novel therapies and have a CPS of 8.5–13 have a high probability for chemotherapy response in the first-line setting. This CPS could be used for risk communication and making decisions with patients, especially in regard to chemotherapy.

## 1. Introduction

Lung cancer is the leading cause of cancer death worldwide and non-small cell lung cancer (NSCLC) is the most common form, accounting for 84% of all diagnoses [1]. Smoking is the major risk factor of all histological types of lung cancer. Unfortunately, most patients present as stage IV disease, which is associated with a poor survival of around 8–10 months with conventional chemotherapy [2]. Targeted therapy, tyrosine kinase inhibitors (TKI), can prolong survival to 40 months, especially in epidermal growth factor receptor (EGFR) gene mutation and anaplastic lymphoma kinase (ALK) rearrangement NSCLC. Recently, immune checkpoint inhibitors (ICIs) have had a greater role in NSCLC with no oncogenic driver mutation. ICIs can replace chemotherapy in the high expression of programmed cell death ligand 1 (PD-L1) and provide durable survival for some patients [3]. Although patients are now surviving longer with novel therapies, most cannot access them due to the cost burden and healthcare reimbursement system [4,5]. Platinum-doublet chemotherapy is still mainly used in the first-line setting especially in resource-limited countries. A fear of side effects and refusal of chemotherapy are problems in clinics [6] and may be mitigated if patients can be reassured about the response before initiating treatment. Unlike TKI and ICI, we have no predictive biomarkers for patient selection prior to chemotherapy treatment. Clinical factors such as younger age, higher body mass index (BMI), and lower Glasgow prognostic score (GPS) have been shown to have better prognosis for overall survival (OS). ERCC1, RRM1, hENT1, and BRCA1 from previous studies showed some potential prediction but are not practical in routine service [7]. Moreover, previous studies that aimed to predict response are limited. This study aims to establish clinicopathologic factors influencing the chemotherapy response in advanced NSCLC patients receiving platinum-doublet chemotherapy in the first-line setting and develop a clinical prediction score to predict the response as a physician’s tool for communication with patients and their families.

## 2. Materials and Methods

This study was designed as a retrospective cohort study with approval by the Research Ethics Committee, Surin Hospital, Ministry of Public Health of Thailand under protocol 23/2562 with an exemption from patient informed consent due to it being a full retrospective study. The study was registered by Thai Clinical Trials Registry number TCTR20220131001.

### 2.1. Participants

The data were collected from the medical record of non-small cell lung cancer (NSCLC) patients with unresectable stage IIIB or IV by TNM 7th edition who were treated with platinum-doublet chemotherapy as a first-line treatment at Surin Hospital Cancer Center between July 2014 and December 2018. Patients older than 18 years who had confirmed cancer by cytopathology and were treated with platinum plus paclitaxel or platinum plus gemcitabine for at least 2 cycles (total 4–6 cycles) were included in this study. Patients who had ECOG 3–4 or planned for radiotherapy were excluded.

### 2.2. Outcome

Computed tomography (CT) examinations were performed using 128 slices per rotation, 1-mm and 5-mm reconstruction in plain and venous phase (Aquilion Prime, Canon Medical Systems, Tustin, CA, USA) or chest radiography was used to evaluate the progression of disease during chemotherapy. The Response Evaluation Criteria in Solid Tumors (RECIST) version 1.1 was used to classify response groups. Patients with a complete response (CR) or partial response (PR) were classified to the responder group and patients with stable disease (SD) or progressive disease (PD) were classified to the non-responder group.

### 2.3. Predictors

Responders had longer progression-free survival (PFS) and overall survival (OS) [7,8,9]. We used parameters that determine survival prognosis due to there being no previous study that used clinicopathologic factors to predict chemotherapy response in the first-line setting of advanced NSCLC. Age and gender were prognostic factors for survival outcome and considered as universal factors [10,11]. Smoking history had a negative impact on chemotherapy treatment [12]. Performance status was measured according to the Eastern Cooperative Oncology Group (ECOG) classification, which ranged from grade 0 (fully active) to grade 5 (dead). ECOG grades 0 and 1 were grouped and had better prognosis from a previous study [13]. Cachexia, defined as significant weight loss of more than 5% within 3 months, had poor survival [14]. Higher body mass index (BMI) was associated with improved PFS and OS [7]. Serum albumin ≥ 3.5 g/dL, one of the factors in GPS, correlated with a good prognosis [13]. A higher white blood cell (WBC) count was correlated with significant worse outcomes in the Prognostic Index (PI) with multiple types of cancer [15]. The serum level of carcinoembryonic antigen (CEA) was found to be a useful prognostic marker for recurrence after surgery, PFS, or OS in NSCLC patients [16].

The parameters recorded in this study were categorized into three groups: general patient information, pre-treatment laboratory data, and pathological finding. The general patient information group consisted of gender, age, smoking status, TNM staging, performance status, history of significant weight loss, and BMI. The pre-treatment laboratory data group included serum albumin, hemoglobin, WBC, absolute neutrophil count (ANC), and CEA. Pathological findings consisted of histologic subtype and tumor grading.

### 2.4. Statistical Analysis

Statistical analysis was performed using the STATA version 16.0 Statistical Package. Data were presented as mean (standard deviation), median (interquartile range), count, or percentage, as appropriate. A Student’s *t*-test or Mann–Whitney U test was used for comparing the continuous variables and a Fisher’s exact test was used for comparing the categorical variables. The significance level was less than 0.05. Factors associated with chemotherapy response were explored by risk regression analysis, reported as risk ratios (RRs) and a 95% confidence interval (CI). Various combinations of significant clinical parameters that had a univariable *p*-value < 0.05 or general clinicians’ concerns were selected in the multivariable model and removed by the stepwise method to retrieve all significant predictive parameters for model development. The parameters that achieved clinical and statistical significance from the multivariable model were assigned the score, which was transformed from the regression coefficient of RR. Age, performance status, and smoking status were included in the model because of general concerns in lung cancer treatment. A total score of 0–13 was derived from the model with a cut-off point of 8.5 to predict a high probability for chemotherapy response. The positive predictive value (PPV) and likelihood ratio (LR) of positive values were presented. The area under the receiver operating characteristics (AuROC) curve was calculated to derive the discriminative potential of the model. Hosmer–Lemeshow goodness of fit statistics and a calibration plot were presented. The bootstrapping procedure with 200 replicates was executed for internal validation of the model.

## 3. Results

A total of 207 NSCLC patients treated with platinum-doublet chemotherapy in the first-line setting were included in this final analysis, as shown in Figure 1. A total of 117 (56.5%) were classified to the responder group and 90 patients (43.5%) to the non-responder group. There were more males (61.1%) in the non-responder group and a greater squamous cell carcinoma histology (33.0%) in the responder group. Additional details are summarized in Table 1. The parameters that showed univariable statistical significance or general clinicians’ concerns were gender, age, smoking status, significant weight loss, BMI, ECOG score, pre-treatment serum albumin, and histologic subtype (Table 1).

Multivariable risk regression was conducted. Six parameters were selected in the model. The regression coefficient, risk ratio (RR), and 95% CI are shown in Table 2. Female gender, age ≥ 60 years, current smoking/ex-smoking status, ECOG score 0–1, pre-treatment serum albumin ≥ 3.5 mg/dL, and squamous cell carcinoma histology were included in the final model. The item scoring for the clinical prediction of responses (chemotherapy response) derived from coefficients is shown in Table 3. The assigned scores ranged from 0 to 13 points, which categorized patients into two risk groups by a cut-off point of 8.5, as presented in Table 4. The score of the low-risk group for being a responder was 0 to 8 points, while 8.5 to 13 points was the score of the high-risk group. The ROC curve and performance of the clinical prediction score model is shown in Figure 2. The AuROC of the model showed a prediction affinity of 0.71 (95% CI 0.63–0.78). The goodness of fit by Hosmer–Lemeshow test of this model was performed with no evidence of lack of fit (*p* = 0.476). The calibration of the model is shown in Figure 3, which compared the observed risk with the score-predicted risk of the responder. A bootstrapping method was performed for internal validation, which showed a consistent AuROC 0.66 (95% CI 0.59–0.73) with model optimism at 0.043 (range from −0.103 to 0.193) and bootstrap shrinkage was 0.776 (95% CI 0.48–1.13).

## 4. Discussion

The results of this retrospective study of 207 advanced NSCLC patients treated with platinum-doublet chemotherapy in the first-line setting with at least two cycles, demonstrate that female gender, squamous cell carcinoma histology, and normal pre-treatment serum albumin are positive predictive factors for chemotherapy response. Responders, patients who achieve an objective response, have longer progression-free survival and overall survival, which has been previously reported [7,8,9]. Bhawna et al. showed that male gender and response with stable disease (SD) after two cycles of platinum-based chemotherapy were independent poor prognostic factors for survival [9] and patient characteristics in this study also had greater squamous cell carcinoma proportions in the partial response (PR) group (*p* = 0.05).

To our knowledge, this is the first study that used clinicopathologic factors in routine practice to predict the response to chemotherapy and develop a clinical prediction score as a simple tool for communication with patients. This model is acceptable for good test quality due to an ROC greater than 0.70. We used a score ranging from 8.5 to 13 for a high probability of being a responder, because a PPV at 83.7% is nearly the response rate of targeted therapies, such as TKI in EGFR-mutated or ALK-rearranged NSCLC [17,18].

In previous studies, the best-validated predictive biomarkers for chemotherapy response were a high expression of ERCC1 (platinum resistance) [19,20] and high expression of RRM1 (gemcitabine resistance) [21], but neither are currently used in clinical implementation [22]. Recently published studies focused on imaging biomarkers, such as contrast-enhanced computerized tomography (CECT) [23], magnetic resonance diffusion-weighted imaging (DWI) [24], and dual-input perfusion CT analysis [25] after two cycles of treatment, are difficult to apply in resource-limited settings.

Moreover, most studies have explored prognostic factors for progression-free survival or overall survival but fewer explored predicting chemotherapy response, which may help more to discuss treatment decisions with patients and their families. Claribel P. Simmons et al. found that performance status combined with a Glasgow prognostic score (GPS) is a superior prognostic factor in advanced lung cancer [13] but their study does not include the histological subtype of NSCLC and includes extensive small cell lung cancer (26.2%), which has a poor prognosis by itself. Concerning performance status, its limitation is subjective assessment. One study conducted by Ahmed Badawy et al. found that smoking, pleural metastases, abdominal metastases, hypoalbuminemia, and hyponatremia are associated with a poor responses to platinum-based doublets [26]. The pre-treatment of serum albumin may be a crucial factor because it is in many prognostic models such as GPS and in our study.

When patients and their families have much concern about chemotherapy toxicity and tend to refuse the treatment, this CPS may be used for a directive purpose. For patients with a high score, clinicians can advise them with confidence to get chemotherapy treatment. In contrast, patients with a low score may be treated appropriately with palliative care. The present study has some limitations. First, most of the patients were unknown for oncogenic driver mutation or PD-L1 status, which may affect chemotherapy response. Second, imaging studies such as CT or MRI cannot be regularly conducted after two cycles under current guidelines due to the healthcare system and, finally, we did not include other platinum-doublet regimens, such as pemetrexed or vinorelbine. The external validation of this clinical prediction score should be performed after this.

## 5. Conclusions

Clinical parameters with predictive factors for being a responder are female gender, a squamous cell carcinoma subtype, and normal pre-treatment serum albumin. Advanced NSCLC patients who cannot access novel therapies and have a clinical prediction score of 8.5 to 13 have a high probability for chemotherapy response in first-line treatment. This CPS could be used for risk communication and making decisions with patients, especially in regard to cytotoxic agents.

## Figures and Tables

**Figure 1 healthcare-11-00293-f001:**
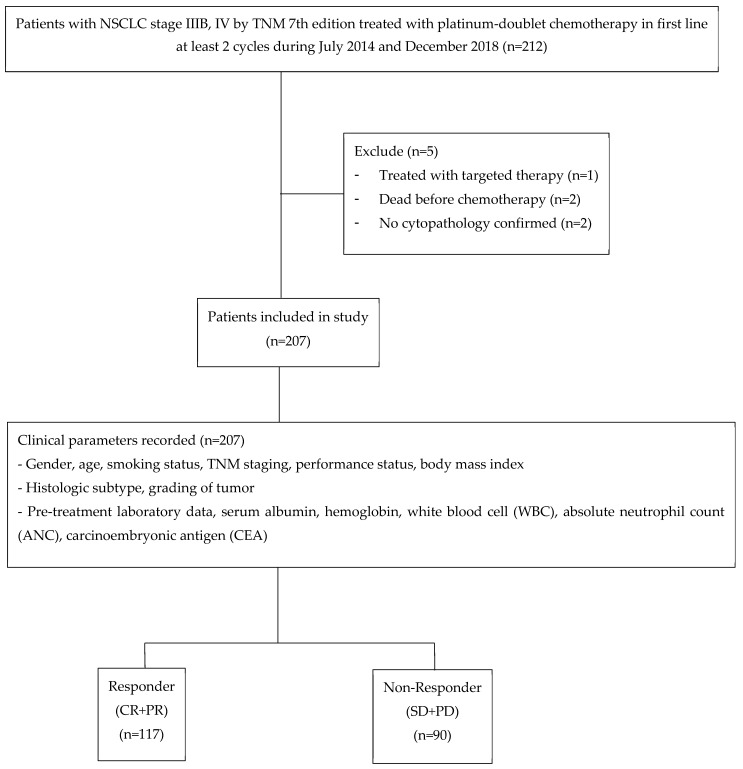
Study flow diagram.

**Figure 2 healthcare-11-00293-f002:**
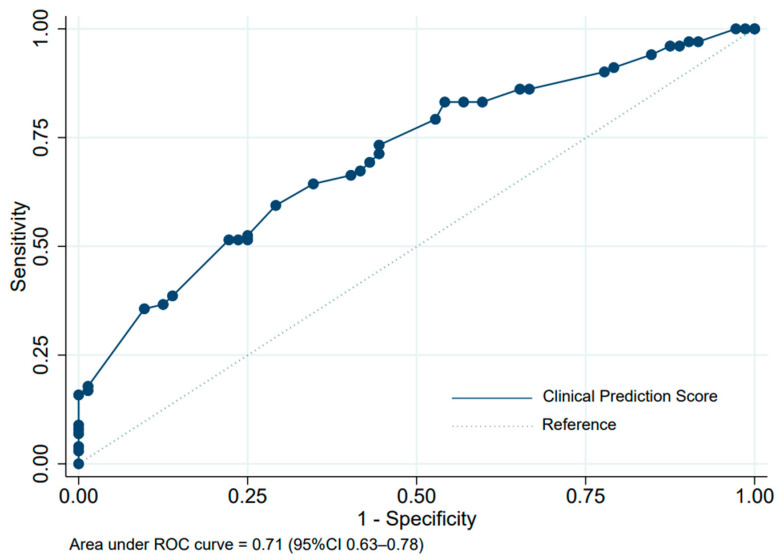
Performance of the clinical prediction score, area under the receiver operating characteristics (ROC) curve.

**Figure 3 healthcare-11-00293-f003:**
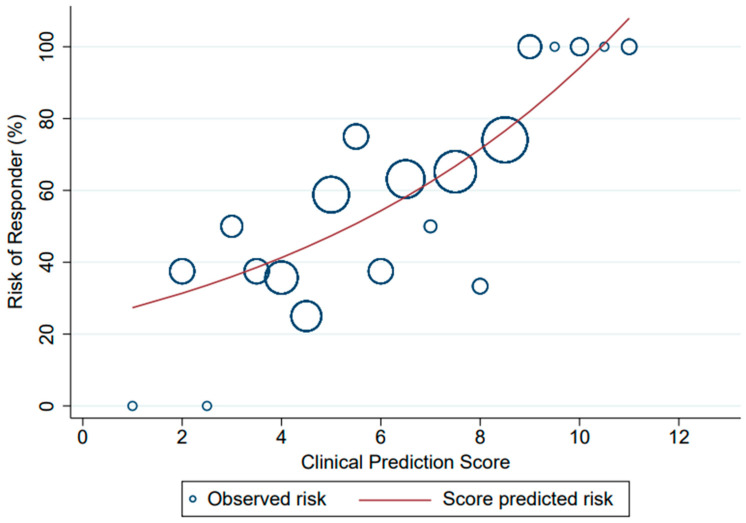
Observed risk (circle) versus score-predicted risk (solid line) of responder probability.

**Table 1 healthcare-11-00293-t001:** Patient factors, pathological factors, and pre-treatment laboratory results of responder and non-responder groups (207 cases).

Characteristics	Responder		Non-Responder		*p*-Value
	CR + PR (*n* = 117)		SD + PD (*n* = 90)		
	*n*	%	*n*	%	
I. Patient factors					
Gender					
Male	50	42.7	55	61.1	0.011
Female	67	57.3	35	38.9	
Age (year) *	62.8	(±9.6)	61.4	(±10.4)	0.298
Smoking status					
Current/ex-smoking	40	34.2	33	38.8	0.554
Never	77	65.8	52	61.2	
TNM staging					
II	1	0.9	0	0.0	0.503
IIIA	3	2.6	2	2.2	
IIIB	8	6.8	4	4.4	
IV	105	89.7	84	93.4	
PS					
ECOG 0–1	102	90.3	70	85.4	0.369
ECOG ≥ 2	11	9.7	12	14.6	
Significant weight loss **					
Yes	35	30.4	28	31.5	0.880
No	80	69.6	61	68.5	
BMI (kg/m^2^)					
<18.5	35	29.9	35	38.9	0.251
18.5–22.9	48	41.0	37	41.1	
≥23	34	29.1	18	20.0	
II. Pathological factors					
Histology					
Adenocarcinoma	72	64.3	62	69.7	0.013
Squamous cell carcinoma	37	33.0	17	19.1	
Large cell carcinoma	0	0.0	1	1.1	
NOS	3	2.7	9	10.1	
Tumor grading					
Well-differentiated	5	4.3	7	7.8	0.209
Moderately differentiated	5	4.3	9	10.0	
Poorly differentiated	21	18.0	13	14.4	
Undifferentiated	0	0.0	1	1.1	
NA	86	73.4	60	66.7	
III. Pre-treatment laboratory results					
Albumin (g/dL) *	3.5	(±0.5)	3.4	(±0.4)	0.069
Hemoglobin (g/dL) *	11.4	(±1.6)	11.5	(±2.3)	0.627
WBCs (cells/mm^3^) *	9737.3	(±4549.8)	9389.8	(±3112.6)	0.536
ANC (cells/mm^3^) *	6617.6	(±3249.3)	6613.9	(±2742.1)	0.993
CEA (ng/mL) ***	47.9	(10.9, 258.4)	42.4	(10.5, 116.2)	0.411

Notes: * mean (±SD), ** 5% within 3 months, *** median (IQR). Abbreviations: PS, performance status; ECOG, Eastern Cooperative Oncology Group; BMI, body mass index; NOS, not otherwise specified; NA, not available; WBCs, white blood cells; ANC, absolute neutrophil count; CEA, carcinoembryonic antigen.

**Table 2 healthcare-11-00293-t002:** Regression coefficient, risk ratio (RR), and 95% CI of selected parameters derived from risk regression after multiple imputation.

Parameters	Coefficient	RR	95% CI of RR	*p*-Value
Female	0.56	1.75	1.20–2.55	0.004
Age ≥ 60 years	0.15	1.16	0.90–1.49	0.259
Current/ex-smoking	0.32	1.38	0.94–2.02	0.100
ECOG 0–1	0.18	1.20	0.75–1.90	0.447
Squamous cell carcinoma	0.35	1.41	1.10–1.81	0.006
Albumin ≥ 3.5 mg/dL	0.33	1.40	1.06–1.85	0.019

**Table 3 healthcare-11-00293-t003:** Item scoring scheme for responder.

Clinical Parameters	Coefficient	Transformed Coefficient	Assigned Score
Gender			
Female	0.56	3.83	4
Male	-	-	0
Age			
≥60 years	0.15	1.00	1
<60 years	-	-	0
Smoking status			
Current/ex-smoking	0.32	2.21	2
Never	-	-	0
ECOG			
0–1	0.18	1.24	1
2	-	-	0
Albumin			
≥3.5 mg/dL	0.33	2.29	2.5
<3.5 mg/dL	-	-	0
Histologic subtype			
Squamous cell carcinoma	0.35	2.37	2.5
Non-squamous cell carcinoma	-	-	0

**Table 4 healthcare-11-00293-t004:** Distribution of risk level for responder, PPV, likelihood ratio of positive (LHR+), and 95% CI.

Risk Level	Responder	Non-Responder	PPV (%)	LHR+	95% CI of LHR+	*p*-Value
	*n* (%)	*n* (%)				
Low (0–8)	65 (50.0)	65 (50.0)	50.0	0.34	0.18–0.66	<0.001
High (8.5–13)	36 (83.7)	7 (16.3)	83.7	3.67	1.73–7.77	<0.001

Abbreviations: PPV, positive predictive value; CI, confidence interval.

## Data Availability

Not applicable.

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
