# Peer review of "Development of Clinical Prediction Score for Chemotherapy Response in Advanced Non-Small Cell Lung Cancer Patients"

_healthcare, 2023, doi:10.3390/healthcare11030293_

Round 1
Reviewer 1 Report
Overall well written. Important tool to help recognize which patients could be poor responders to treatment and also aide in communication. Although it is a better tool as it uses clinical and lab parameters, important to assess if there is overall survival correlation to response to treatment. More studies are needed in that regard.
Author Response
Point 1: Although it is a better tool as it uses clinical and lab parameters, important to assess if there is overall survival correlation to response to treatment. More studies are needed in that regard.
Response 1: Thank you for your comments and suggestions. There were some studies showed response rate of first line chemotherapy in advanced NSCLC as a surrogate marker for overall survival. Nakashima et al. (JTO 2016) conducted systematic review including 44 eligible articles evaluated chemonaive patients with advanced NSCLC, found RR, followed by PFS had the strongest association with overall survival but not strong enough to replace it. Recently, we presented a poster with prognostic indicators for chemotherapy response in advanced NSCLC in World Conference on Lung Cancer 2021. Median OS of patients with response was 12.5 months (95%CI,11.0-13.6) and patients with SD or PD (non-responder) was 8.3 months (95%CI,7.0-10.3), HR 0.56, p=0.001 by log-rank test.
https://pubmed.ncbi.nlm.nih.gov/27178983/
https://www.jto.org/article/S1556-0864(21)02884-7/fulltext
Reviewer 2 Report
Dear authors,
I was honored to review your manuscript.
The paper is well written. The methods and the results are well described.
I would recommend to expand your introduction chapter. For example, you mention that "Lung cancer is the leading cause of cancer death worldwide and non-small cell lung cancer (NSCLC) is the most common form, accounting for 84% of all diagnoses..."
I would have mentioned the risk factors for developing this type of cancer, and some survival data with the novel therapies that you have mentioned later on.
Author Response
Point 1: I would have mentioned the risk factors for developing this type of cancer, and some survival data with the novel therapies that you have mentioned later on.
Response 1: Thank you for your suggestions. I will revise my manuscript as your comments.
Reviewer 3 Report
The authors have taken on an important issue, the ability to predict response to chemotherapy in advanced NSLC patients, to develop a scoring algorithm that would be easy to apply
I find the goal to be laudable but have questions about the selection of parameters that were included in the analysis
1. why were these specific parameters selected? the discussion separates them into 3 major categories but does not address their selection
2. several of the parameters are themselves "composite" parameters, e.g. even TMN staging as there are different TMN scores that can be associated with the same stage; similarly smoking categorization would be improved if the category of current/ex-smoking were resolved in terms of current, within last two years, etc; and weight loss of 5% in the last year could be better defined in terms of whether the patient might have been exercising/dieting to accomplish this or it is more directly related to disease
Author Response
Thank you for your comments and suggestions.
Point 1: why were these specific parameters selected? the discussion separates them into 3 major categories but does not address their selection
Response 1:
First category (general information):
-Gender and age are both universal factors.
-Smoking status is considered a factor because of different in genetic background in some subtype of lung cancer such as we can found oncogenic-driven mutation in squamous cell carcinoma who is non-smoker, that may affect with prognosis and treatment response.
-TNM staging and performance status (ECOG) both are prognostic factors in general cancer patients.
-BMI at cancer diagnosis, it is association with survival in many type of cancers.
Second category (pre-treatment lab):
-Serum albumin is one factor in validated mGPS (modified Glasgow Prognostic Scores), this score is used to predict cancer outcome. We do not include CRP because it is not considered as basic lab in resource-limited.
-CEA is also prognostic marker in NSCLC.
Third category (pathological finding):
-Prognosis is depend on subtype and grading of lung cancer.
Point 2: several of the parameters are themselves "composite" parameters, e.g. even TMN staging as there are different TMN scores that can be associated with the same stage; similarly smoking categorization would be improved if the category of current/ex-smoking were resolved in terms of current, within last two years, etc; and weight loss of 5% in the last year could be better defined in terms of whether the patient might have been exercising/dieting to accomplish this or it is more directly related to disease
Response 2: Thank you for your suggestions. Smoking status, we general used definition of NHIS, but limitation of retrospective study, there may be loss some data and misunderstand with other physicians. As many oncology trials, smoking status was categorized into current/ex-smoking and non-smoking.
We used definition of significant weight loss (in 3-6 months) to determine prognosis of cancer that correlated with overall survival.
https://www.ncbi.nlm.nih.gov/pmc/articles/PMC2409471/
Round 2
Reviewer 3 Report
the authors provided responses to my original review questions but only directed outside the publication. I strongly urge the authors to expand upon their decisions regarding these parameters to be included in the article...
I am not convinced yet that they have made adequate choices about their parameter selection nor validated them so believe that it is critical that greater transparency be included so that readers can better evaluate the results themselves
Author Response
Point 1: the authors provided responses to my original review questions but only directed outside the publication. I strongly urge the authors to expand upon their decisions regarding these parameters to be included in the article...
I am not convinced yet that they have made adequate choices about their parameter selection nor validated them so believe that it is critical that greater transparency be included so that readers can better evaluate the results themselves
Response 1: I've revised the latest manuscript and add about parameters that be choosed in part of predictors (Method). Thank you for your comments.
Round 3
Reviewer 3 Report
the authors have addressed my comments